# Digital Assessment of the Knowledge, Attitudes and Preparedness of Dentists towards Providing Dental Treatment to People Living with HIV in Northern Brazil

**DOI:** 10.3390/ijerph20196847

**Published:** 2023-09-27

**Authors:** Ricardo Roberto de Souza Fonseca, Rogério Valois Laurentino, Silvio Augusto Fernandes de Menezes, Aldemir Branco Oliveira-Filho, Paula Cristina Rodrigues Frade, Roberta Pimentel de Oliveira, Luiz Fernando Almeida Machado

**Affiliations:** 1Biology of Infectious and Parasitic Agents Post-Graduate Program, Federal University of Para, Belem 66075-110, PA, Brazil; ricardofonseca285@gmail.com; 2Virology Laboratory, Institute of Biological Sciences, Federal University of Para, Belem 66075-110, PA, Brazil; valois@ufpa.br; 3School of Dentistry, University Center of State of Para, Belem 66060-575, PA, Brazil; menezesperio@gmail.com; 4Study and Research Group on Vulnerable Populations, Institute for Coastal Studies, Federal University of Para, Belem 68600-000, PA, Brazil; olivfilho@ufpa.br; 5Tropical Medicine Nucleus, Federal University of Para, Belem 66075-110, PA, Brazil; paulacrfrade@gmail.com; 6School of Dentistry, Federal University of Para, Belem 66075-110, PA, Brazil; betapimentel@hotmail.com

**Keywords:** dental education, health knowledge, dentistry, HIV, stigmatization, PLWHA

## Abstract

People living with HIV (PLWH) continue to face stigma and discrimination during dental treatment in Brazil. This study aimed to describe the sociodemographic, educational and professional characteristics of dentists working in the Northern Brazilian region and to identify the level of knowledge of these health professionals in relation to the care and management of PLWH using a digital form. Methods: This cross-sectional study was population-based among dentists and was conducted between July and December 2021. A total of 396 dentists were invited using the “snowball” sampling technique and received an anonymous digital form (Google^®^ Forms platform) composed of four blocks of dichotomous and multiple-choice questions. After signing the informed consent form, dentists were divided into seven groups according to the amount of time that had passed (in years) since they had completed their bachelor’s degree in dentistry. A total of 25 questions referring to sociodemographic, educational and professional factors and knowledge about the dental care of PLWH were asked, and all data were presented as descriptive percentages and then analyzed using the Kappa test. Results: The average age was 31.9 years, and the states of Pará and Amazonas presented better knowledge about the dental care of PLWH. Dental prostheses (62/381-16.3%), orthodontics (57/381-15%) and periodontics (56/381-14.7%) were the most cited dental specialties, and demographic, professional and epidemiological characteristics showed a statistically significant difference regarding knowledge (<0.0001). Conclusions: The knowledge of dentists in Northern Brazil is partially high, although there is some lack of knowledge about the dental care of PLWH.

## 1. Introduction

In Brazil, public or private health services’ availability versus population access to health services has always been heterogeneous, mainly due to regional inequalities, such as social, cultural, economic, demographic and geographic inequalities, and its resolution is a complex challenge for the Brazilian government [1]. Specifically, there are differences between health service availability and population access because, in some regions, such as the northern region of Brazil, the availability of public and/or private health services, such as dentistry, to the population may be compromised mainly by economic and geographic barriers, which directly impact dental health, decreasing quality of life, especially among key populations infected with HIV [2,3].

According to Fonseca et al. [4], several oral manifestations, such as oral candidiasis, necrotizing gingivitis or periodontitis, Kaposi sarcoma and leukoplakia, are commonly found among PLWH, serving as diagnoses or indications for HIV infection or determining the individual’s stage of HIV infection, demonstrating the major importance of dentists in specialized AIDS treatment and HIV infection control [5]. Unfortunately, some studies have reported certain unwillingness by dentists and dental students to properly treat PLWH, which has been associated with stigma about HIV transmission during dental management or inadequate knowledge of the HIV infection process, resulting in fear, phobia or stigma towards HIV and contamination, demonstrating that dentists and dental students require specific education and training in specialized AIDS treatment and HIV infection control by universities, health institutes and the federal government [6,7,8].

Historically, HIV infection and key populations are strongly marked by stigma and prejudice by families, society and even health workers, such as dentists, which may lead affected individuals to experience social isolation and physical, dental and mental health problems [9]. Interestingly, regarding dental care or treatment, previous studies demonstrated that PLWH were unwilling to seek dental treatment and hesitated to disclose their HIV status to dentists as a way to prevent stigmatization. In the literature, it has been reported that dentists with low knowledge levels about HIV infection often refuse to treat PLWH, and a lack of knowledge in dental workers, dental students and dentists can decrease oral health and quality of life among PLWH due to the higher presence of oral diseases such as periodontitis [10,11,12].

Worldwide, a diversity of forms in different studies have been used to measure the knowledge, attitudes and willingness towards the dental treatment of PLWH [13,14,15,16]. Despite different evaluation methods, the majority of studies showed that the lower willingness to treat HIV patients among dental students or dentists might be linked to their knowledge about HIV infection and disease aspects, the recognition of oral manifestations and the understanding of the modes of transmission. These results may indicate that HIV/AIDS stigma among dentists and dental students is directly related to lower knowledge levels, which leads to higher levels of stigma or unwillingness to treat affected individuals [15,16].

Therefore, this lack of knowledge, fear of contracting HIV and HIV/AIDS multidisciplinary treatment remain a major dental health problem for individuals affected by HIV or key populations in countries such as China, India, Indonesia and Brazil, mainly in the northern region of Brazil [17]. Thus, this study describes the sociodemographic, educational and professional characteristics of dentists working in the Northern Brazilian region and identifies the level of knowledge of these health professionals in relation to the care and management of PLWH using a digital form and as knowledge improvement a booklet is used as Appendix A.

## 2. Materials and Methods

### 2.1. Region Characterization, Study Design and Sample Size

The northern region of Brazil is composed of seven states: Acre (capital: Rio Branco), Amapá (capital: Macapá), Amazonas (capital: Manaus), Pará (capital: Belém), Rondônia (capital: Porto Velho), Roraima (capital: Boa Vista) and Tocantins (capital: Palmas) [7,18]. The northern region is characterized by a large area of the Amazon rainforest and a variety of watersheds that represent a territorial area of 3,853,676 km^2^, which is almost 45% of the Brazilian national territory, with a total population of 18,906,962 people, making the northern region the least inhabited region in Brazil. In addition to geographical isolation, the northern region of Brazil has a highway connectivity limitation, lower industrialization areas, decreased goods productivity, higher poverty rates and less structured public health services, causing difficulties in accessing specialized AIDS treatment [18,19,20].

This electronic, descriptive and cross-sectional single-center study was population-based and studied dentists who were currently actively registered in the Brazilian Federal Council of Dentistry (CFO) and residing in Northern Brazilian states during the period from October 2021 to October 2022. The sample size was calculated based on the population of 19,161 dentists with active registration in 2021 at CFO. Using a 95% confidence level, 5% sampling error and considering a 5% margin of error, there was the need for 378 dentists to respond to the digital form among all 7 states (Figure 1).

### 2.2. Ethics

The present study was approved by the Research Ethics Committee of the Institute of Biological Sciences at the Federal University of Pará (UFPA) under protocol number 4,606,188. Informed consent was obtained from all the participants that were included in the analyses in this study.

### 2.3. Data Collection

A nonprobabilistic “snowball” sampling technique was used to invite dentists to participate in the present study [21]. Initially, six dentists from each Northern Brazilian state were contacted by the authors, as indicated by the Regional Council of Dentistry (CRO). These six dentists received detailed information on the objectives of the study, the data collection procedures and data security and were asked to sign written informed consent forms before participating in the study [20].

Then, they were requested to help publicize the study and invite other dentists to participate. The digital form used to collect data was distributed using WhatsApp, Facebook, Instagram (Facebook Inc., Menlo Park, CA, USA) and Telegram (Telegram Messenger LLP, Moscow, Russia). Following this, each dentist received and filled out their own digital form, and then they were asked to invite three other dentists to participate. This procedure was repeated a number of times until the study sample was complete [22].

All participants were informed of the nature of the study and its potential risks and benefits after they were required to sign the digital written informed consent form and before being given access to the digital form. The inclusion criteria were as follows: (i) ≥22 years old, (ii) complete graduation in a CFO-registered dentistry course, (iii) active registration in the CFO and CRO, (iv) based in any Northern Brazilian state and (v) access to the internet. Potential participants who did not sign the informed consent form or did not have stable internet access were excluded from the study. The exclusion criteria were as follows: (i) individuals who moved to other states during the study, (ii) dental students or undergraduates, (iii) those with no access to the internet and (iv) those with any type of impairment that prevented them from completing the digital form [20].

The dentists were divided into seven subgroups according the amount of time that had passed (in years) since they had completed their bachelor’s degree in dentistry, and the groups were organized as follows: G1: ≤1 year (control group); G2: 1 year; G3: 2 years; G4: 3 years; G5: 4 years; G6: 5 years; and G7: ≥5 years. This division was applied because we aimed to understand whether more experienced dentists had more knowledge than less experienced dentists and to understand the connection between the number of years of dental clinical experience and HIV knowledge among dentists.

### 2.4. Digital Form

A digital form was used to collect data. This form was formatted and administered using the Google^®^ Forms platform (Mountain View, CA, USA), and it was distributed electronically through the social media platforms mentioned above. The digital form was composed of four question blocks, and the participants could not advance to the next block without filling in all the mandatory questions in the current block [20]. This digital form was previously used and validated in a pilot study published in 2022 [20].

Blocks 1 and 2 contained information on the researchers, study objectives, risks and potential benefits, as well as the consent form. Block 3 included 11 questions about sociodemographic, educational and professional characteristics such as (i) age, (ii) gender, (iii) current source, (iv) amount of time that had passed (in years) since they had completed their bachelor’s degree in dentistry, (v) the college institution type (public or private), (vi) whether they had a postgraduate qualification (sensu stricto or sensu lato), (vii) whether they had any dental specialty, (viii) the type of workplace (public or private), (ix) whether they had already participated in any class about HIV or sexually transmitted infections (STIs), (x) whether they had already provided oral care to PLWH and (xi) whether they knew how to proceed after a work accident.

Block 4 contained 12 questions regarding technical–scientific knowledge regarding HIV infection, transmission, diagnosis, clinical signs and symptoms, common oral manifestations, the clinical and dental management of PLWH, biosafety and how to proceed during work accidents [20]. Additionally, an extra section was available to individuals willing provide self-responses or histories or describe their fears or stigmas about the clinical and dental management of PLWH. Finally, a scale from 0 to 10 was used at the beginning and end of the digital form to measure the self-reported knowledge of dentists before and after completing the digital form. For analysis, a self-reported score from 0 to 6 was considered insufficient knowledge, and a self-reported score from 7 to 10 was accepted as sufficient knowledge [20].

### 2.5. Statistical Analysis

All the data collected were entered into an Excel database (Microsoft Corp., Redmond, WA, USA), and then all statistical procedures were run in the BioEstat 5.0 program (Informer Technologies Inc., Manaus, Brazil). Absolute and relative frequencies, means, medians, amplitudes and standard deviations were used to describe dentists’ sociodemographic, educational and professional characteristics in terms of the quantitative and qualitative variables using chi-square and G tests. The dentists’ technical–scientific knowledge level was assessed using two approaches. The first was to compare the self-reported knowledge in blocks’ 3 and 4’s answers using the Kappa and confidence interval (CI) between groups.

The second evaluation compared the self-reported technical–scientific knowledge level scores from the scales of the digital form. The difference was evaluated using the Kappa test, which verified the significance of the possible difference between the first and second responses (based on a *p* < 0.05 significance level).

## 3. Results

### 3.1. Sample

In total, 396 dentists participated in this study, of whom 42 dentists were initially contacted, with six dentists from each Northern Brazilian state, according to the “snowball” sampling method. From the first six dentists, the other 354 dentists were contacted, and of these 354, 15 (3.8%) were excluded from this study: 10 dentists were excluded due to not signing the consent form, three dentists were not working in northern states and two had no active registration in the CFO and CRO at the time of completing the digital form. Therefore, the final sample for this study consisted of 381 (96.2%) dentists working in Northern Brazilian states, which were divided into the following groups: G1 (control group—recent graduates): 51/381 (13.4%); G2: 28/381 (7.3%); G3: 37/381 (9.7%); G4: 46/381 (12.1%); G5: 35/381 (9.2%); G6: 28/381 (7.3%); and G7: 156/381 (40.9%).

### 3.2. Professional, Demographic and Epidemiological Characteristics

The demographic, professional and epidemiological characteristics of the 381 dentists are shown in Table 1. The sample was predominantly composed of 193/381 female dentists (50.7%), while 188/381 (49.3%) were male dentists. The average age was 31.9 years (range 22 to ≥59). The majority of dentists worked in Pará (140/381, 36.7%) and Amazonas (41/381, 10.8%), which are territorially the largest states in the northern region and have the highest rates of registered and active dentists according to the CRO. Among the groups and states, in Pará, G7 (62/381, 39.7%) and G1 (29/381, 56.8%) were the most prevalent, although, in Amazonas, G7 (16/381, 10.2%) and G4 (7/381, 15.2%) were the most prevalent. As shown in Table 1, almost all parameters evaluated had statistical significance (<0.0001), except gender, which had no statistical significance (0.1787).

If dentists had completed a postgraduate qualification, 273/381 (71.7%) had a specialization degree, followed by 94/381 (24.7%) who had not completed their first or other postgraduate degrees, 77/381 (20.2%) who had a master’s degree and 28/381 (7.3%) who had a doctoral degree. Regarding dentistry specialization, the top five were dental prosthesis: 62/381 (16.3%); orthodontics: 57/381 (15%); periodontics: 56/381 (14.7%); implantology: 55/381 (14.4%); and endodontics: 44/381 (11.5%). Additionally, 110/381 (28.9%) dentists claimed to be participating in an ongoing dentistry specialization.

Regarding dentists’ workplaces, which was a multiple-choice query, 273/381 (71.7%) stated that they worked at a private workplace, and 167/381 (43.8%) stated that they worked at a public workplace. With reference to previous dental care and management experiences of PLWH, dentists were asked if they had already provided dental care to PLWH, and, among the dichotomous query answers, the answers were yes, no and maybe (this last option was included because, in Brazil, many PLWH conceal their HIV infection status due to social and cultural stigma). In total, 225/381 (59.1%) answered yes, 89/381 (23.3%) answered maybe and 67/381 (17.6%) answered no.

The number of yes answers by group was G7: 129/225 (82.7%); G4: 27/225 (58.7%); and G6: 22/225 (78.5%). The number of maybe answers by group was G1: 21/89 (58.8%); G3: 15/89 (40.6%); and G4: 14/89 (30.4%). The number of no answers by group was G1: 20/67 (39.2%); G5: 13/67 (37.1%); and G7: 13/67 (8.3%). The final query of block 3 asked whether dentists had ever participated in any kind of class or lecture about PLWH dental care; 210/381 (55.1%) answered yes and 171/381 (44.9%) answered no. The yes answers per group were G7: 108/210 (69.3%); G1: 27/210 (53%); and G4: 19/210 (41.3%). The no answers per group were G7: 48/171 (30.7%); G5: 28/171 (80%); and G4: 27/171 (58.7%).

### 3.3. Self-Reported Knowledge about HIV and PLWH Dental Care

The fourth query block was about dentists’ self-reported knowledge of HIV concepts, infection mechanisms, transmission pathways, window period and common systemic/oral lesions in PLWH, but also referred to ART medications, laboratory exams and the correct dental protocols to carry out in the dental management of PLWH. According to their answers, they were considered apt or inapt based on the known literature (Table 2). These queries were used because, in theory, all health professionals should possess knowledge of at least these topics regarding HIV and its infection so that HIV treatment can be performed properly. Therefore, according to our results, we can infer that the participating dentists need to improve their knowledge about HIV and its infection. As seen in Table 3, we found statistical significance (<0.0001) among queries 4, 6, 7, 8, 10, 11 and 12.

The first query was whether the dentists knew what HIV and AIDS were; 276/381 (72.4%) answered no/inapt. Per group, these answers were G7: 118/276 (75.7%); G4: 33/276 (71.7%); and G3: 31/276 (83.7%). This demonstrates a lack of basic knowledge about HIV. The second query was whether dentists knew how the HIV infection mechanism worked: 211/381 (55.4%) answered no/inapt. Per group, these answers were G7: 77/211 (49.3%); G1: 30/211 (58.9%); and G5: 26/211 (74.2%).

Queries 3, 4 and 5 were multiple-choice queries. Query 3 was about HIV transmission pathways and consisted of five possible correct answers (vertical transmission, contamination by work accidents, contaminated blood transfusion, unprotected sex and sharing needles to inject drugs) and five possible incorrect answers (sharing hygiene products, aerosol transmission during dental care, personal contact by handshake or hug, sharing plates and cutlery and contact with mucous membranes, wounds or body fluids). The most prevalent answers were unprotected sex: 356/381 (93.4%); contamination by work accidents: 354/381 (92.9%); contaminated blood transfusion: 353/381 (92.7%); sharing needles to inject drugs: 343/381 (90%); and vertical transmission: 262/381 (68.8%). Among incorrect answers, one item caught our attention as the most prevalent answer: 136/381 (35.7%) answered that aerosol transmission during dental care was an HIV transmission pathway (the prevalence of this answer might have been mainly due to the COVID-19 pandemic).

Query 4 was about the most common HIV oral lesions in PLWH. It consisted of all 13 of the most common oral lesions, namely xerostomia, aphthous ulcerations, condyloma, necrotizing periodontitis, oral candidiasis, herpetic gingivostomatitis, necrotizing gingivitis, hairy leukoplakia, linear gingival erythema, necrotizing stomatitis, focal epithelial hyperplasia, Kaposi’s sarcoma and angular cheilitis, as a possible answer, and the correct answer encompassed all of the above; only 114/381 (29.9%) answered correctly. Per group, the answers were G7: 72/156 (46.1%); G6: 12/28 (42.8%); G4: 12/46 (26%). Therefore, the more experienced dentists knew all oral lesions associated with HIV better than less experienced dentists, as measured by the amount of time passed (in years) since they completed their bachelor’s degree in dentistry.

Query 5 was about the most common signs and symptoms of HIV infection, such as prolonged flu, fever, dry cough, malaise, sore throat, nausea, vomiting, lymph node enlargement, fatigue, constant weight loss, night sweats, persistent diarrhea and appetite loss. Like query 4, all of the abovementioned signs and symptoms are common during all stages of HIV infection, so the correct answer again encompassed all of them. Here, 138/381 (36.2%) answered correctly, 81/156 (52%) in G7, 16/46 (34.7%) in G4, and 12/28 (42.8%) in G6. Comparing the answers to queries 4 and 5, the dentists demonstrated better knowledge of the common signs and symptoms of HIV infection than common oral lesions.

Query 6 was about how to request standard HIV medical laboratory exams such as viral load or CD4 cell count tests, and 208/381 (54.6%) answered incorrectly. Query 7 was about whether dentists knew how to proceed after occupational exposure accidents during the dental care of PLWH, and 195/381 (51.2%) answered incorrectly. Query 8 was about whether dentists knew what the window period was, and 197/381 (51.7%) answered correctly. Query 8 was about whether dentists would change dental materials and instrument sterilization standard protocols if they knew that the patient had HIV, and 251/381 (65.9%) answered no. Query 9 was about whether dentists knew that ART medications were used in Brazil with any kind of drug interactions with dental medications, and 216/381 (56.7%) answered no.

Queries 10 and 11 were about whether dentists changed personal protective equipment biosafety standard protocols or were more careful during PLWH’s dental management. Query 10 was about possible changes during noninvasive procedures such as restorations, and 235/381 (61.7%) answered incorrectly. Per group, the answers were G7: 130/156 (83.3%); G5: 22/35 (62%); and G4: 20/46 (43.5%). Query 11 was about these possible changes during invasive procedures such as dental extraction, and 246/381 (64.6%) answered incorrectly. Per group, these answers were: G7: 125/156 (80.1%); G5: 29/46 (63%); and G4: 22/35 (62%). The final query was about whether dentists had any kind of fear or stigma regarding PLWH dental care, and 295/381 (77.4%) answered yes.

### 3.4. Self-Reported Scaling Knowledge

Table 3 and Table 4 represent the analysis of a scale set at the beginning and end of the digital form to evaluate each dentist individual’s perception of their self-reported knowledge at two different times. This scale was included because, after filling out the digital form, all dentists received their answers to see if they were apt or inapt in each category, and if the answer was considered inapt, there was an explanation based on the literature as to why it was considered inapt and how to fix this knowledge; therefore, this digital form was also an instrument to improve participants’ self-reported knowledge. This scale was based on Fonseca et al. [20], so, from 1 to 5, self-declared knowledge was considered insufficient knowledge to care for PLWH, and from 6 to 10, their self-declared knowledge was considered sufficient. As demonstrated in our results, approximately 102 (11.8%) individuals improved their knowledge from insufficient to sufficient. Additionally, the overall self-reported knowledge improved even if dentists had declared sufficient knowledge at the beginning and end.

According to the agreement kappa test used for the scales placed at the beginning and the end of the questionnaire after all queries, we found statistical significance (<0.0001), which suggests that, after answering, all participants showed a knowledge improvement about the topic addressed, which was reinforced by the knowledge variation before and after answering the queries, as demonstrated in Table 4.

## 4. Discussion

This cross-sectional digital study was based on the pilot study of Fonseca et al. [20], which sought to assess the knowledge level of dentists about PLWH oral care only in Pará, Northern Brazil. As stated by the pilot study, the topic of the knowledge level about PLWH is no longer a novelty in Brazil and worldwide. To the best of the authors’ knowledge, this is the first study to investigate this knowledge level on a large scale in Northern Brazil among the dentist population alone. The focus on dentists was due to their importance in HIV infection control or treatment and in diagnosis or currently even in HIV saliva screening tests, so dentists are one of the main forefronts in HIV/AIDS detection, diagnosis and treatment. According to Silva et al. [23] and Santella et al. [24], even at the early stages of infection, some particular oral signs and symptoms, such as candidiasis, oral hairy leukoplakia and Kaposi’s sarcoma, could be seen by dentists and they could alert patients and healthcare services, so that patients can seek preventive or treatment services.

Unfortunately, at present, PLWH experience a range of barriers related to HIV stigma even among health professionals, dentists included. HIV-related stigma is present worldwide but is especially prevalent in developing regions such as Northern Brazil. According to UNAIDS [25], stigma, social discrediting and discrimination against PLWH are commonplace in the Brazilian Amazon region, mainly against key populations such as men who have sex with men, transgender people, sex workers, people who use drugs, people in prison and detention facilities [26,27] and indigenous populations [28]. This HIV-related stigma issue is also important because stigmatizing attitudes and behaviors, mostly towards vulnerable key populations, will directly decrease progress in identifying, diagnosing and treating PLWH, which will affect their quality of life and infection morbidity.

HIV-related stigma and discrimination also impact interpersonal relationships, and, specifically in healthcare advancement, Judgeo and Moalusi [29] defined HIV-related stigma as a particular kind of relationship between an attribute or stereotype or mistaken information that is deeply discredited by the individual. Among healthcare professionals, such dentists’ HIV-related stigma is mainly related to a lack of knowledge and specific training, which leads to fears or doubts regarding the oral care of PLWH. Although this problem has been identified in the literature, no studies have explored the fears or doubts of dentists regarding the oral care of PLWH or tried to solve this issue, which is why this study is novel in the literature, and it presents some additional and fundamental information to address HIV-related stigma.

According to our results, graduated dentists demonstrated a low knowledge level regarding PLWH dental management and, in some cases, a certain stigmatization during PLWH treatment. These results support the idea presented in the worldwide literature of a lack of knowledge about the HIV infection course, oral and systemic manifestations, dental biosafety, ART and transmission pathways [30]. Previous published studies showed that HIV stigmatization among dental students and their unwillingness to treat PLWH are influenced by a lack of knowledge and dental occupation experience, which corroborates our results, because it has been demonstrated that most experienced dentists in our study presented more knowledge than those dentists who had recently graduated or had less dental occupation experience.

Keser et al. [30] evaluated the knowledge level among 200 dental students about HIV, oral signs and attitudes towards HIV infection in Turkey. The authors stated that the knowledge level was low in fourth- and fifth-grade dental students, and, according to Keser et al. [30], participants agreed that PLWH dental treatment poses a higher risk of HIV transmission to dentists than other pathways of HIV transmission, which demonstrates a lack of knowledge and creates explicit stigma against HIV. Additionally, the study showed that the greater the dental student’s grade level was, the higher their knowledge about HIV/AIDS. Therefore, dental students, dentists and dental practitioners should increase their knowledge about HIV/AIDS and PLWH dental care to decrease stigma and increase their willingness to treat individuals affected by HIV, as this value was 29.5% in the Keser et al. [30] study.

When studies have analyzed knowledge levels about HIV and dental treatment, dental students have usually participated, and these studies evidence that dental students from developing countries have low knowledge levels about HIV in addition to stigma and discrimination. Nasir et al. [31] evaluated 642 dental students attending the third, fourth and fifth grades in Sudan regarding HIV and AIDS-related knowledge and stated that 50% reported a need for further education on HIV/AIDS. The authors concluded that dental students attending private dental colleges had better knowledge about HIV than students attending public colleges, which corroborates our results, although, in our sample, 215/381 (56.4%) attended private dental colleges and 166/381 (43.6%) attended public dental colleges in Northern Brazil.

In 2014, Vijayalaxmi et al. [32] conducted a study with 390 dental healthcare professionals and students in India about their willingness to provide dental treatment to PLWH, and, according to the authors, postgraduate students had higher (82.5%) willingness to treat PLWH than undergraduate students. The authors’ results corroborate our results that as one’s dental occupation experience increases, knowledge increases. The self-informed knowledge scale in G1 was 16 (15.8%) and that in G7 was 122 (39.1%). In 2022, Wimardhani et al. [33] evaluated, among 1280 Indonesian dental students, knowledge about HIV transmission, oral manifestation, attitudes towards PLWH and willingness to treat PLWH, and only 63% of Indonesian dental students scored higher than 70% for knowledge of HIV/AIDS.

Rungsiyanont et al. [34] investigated the knowledge and attitudes of 446 dental students, dental practitioners and dentists in Thailand regarding PLWH. Of the 446 participants, 11.9% were dental students, 29.1% were general dentists, 15.5% were specialist dentists, 30.5% were dental hygienists and 13% were dental assistants. The participants had 80% knowledge about HIV subjects, such as transmission pathways and common opportunistic infections. However, knowledge decreased for questions regarding HIV pathogenesis and clinical complications, and 20.4% said that they would deny dental treatment to PLWH if they knew that patients were affected by HIV. To explain how stigma and discrimination might affect dental students, dentists and dental practitioners, Philip et al. [35] evaluated the knowledge levels of HIV 339 healthcare students from Trinidad and Tobago. The authors described that participants had a low knowledge level about HIV; Philip et al. [35] stated that a cognitive–affective factor might be relevant, in addition to a lack of knowledge of stigma against HIV and key populations.

In the current literature, almost all studies worldwide show a lack of knowledge about the dental care of PLWH and show stigma around the treatment of key populations affected by HIV [30,31,32,33,34,35,36,37]. Additionally, in Brazil, in the southern region, studies [38,39] have demonstrated similar results to others, although a bias might be present in these studies because they were all executed in only one region or city. Our studies were the first in the literature to use a questionnaire in all seven Northern Brazilian states, which gave our results increased plausibility and showed that the states of Pará and Amazonas had better overall knowledge about the dental care of PLWH than other states. Additionally, it was demonstrated that periodontics and maxillofacial surgery had greater knowledge levels than other dental specialties, such as orthodontics or dental prothesis. Therefore, more studies in other Brazilian regions must be conducted to understand the particularities of each region and implement student and dental public health service reforms appropriate to each location.

Regarding HIV transmission pathways, it is well established in the literature that apart from direct exposure to contaminated blood, other body fluids, such as semen, vaginal secretions, breast milk, amniotic fluids, sweat, tears, urine and saliva, are considered potentially infectious pathways, similar to unprotected sexual relations, occupational exposure, sharing needles or syringes with injected drugs, perinatal transmission and blood transfusions [40]. In our questionnaire, other non-transmission pathways, such as via dental aerosols, were included to demonstrate whether the participants were aware of the correct HIV transmission pathways, and our results demonstrated that the majority of dentists knew about the correct HIV transmission pathways.

It is well documented that oral lesions may be the first signs of HIV infection in undiagnosed patients. According to the study by Singh et al. [41], knowledge about oral lesions was significantly higher among dental students; unfortunately, our results among professional dentists demonstrated that only 114/381 (29.9%) participants knew about all oral lesions, which is a major concern, especially among G7 participants. Another key point noted in Table 3 regards HIV medications, mainly because the majority of participants were interested in knowing about preexposure prophylaxis (PrEP) and post-exposure prophylaxis (PEP), as 216/381 (56.7%) were unaware of HIV medications. As stated by Kasat et al. [40], these medications are used when an individual knows that he or she will be exposed to HIV, which is why PrEP is used to prevent any contamination or is used by a person who has already been exposed to HIV, ensuring that systemic infection does not occur immediately. Therefore, the use of PEP as soon as possible may prevent systemic infection [40].

While this larger study was successful in terms of the data collected and present data, it had certain limitations, such as its restriction to the Northern Brazilian region, showing only northern characteristics. The exclusion of individuals who did not have access to the internet or did not wish to participate in our study may introduce a bias into our results. Additionally, to complete the digital form quickly, participants may have answered quickly or may not have disclosed the correct demographic information. To prevent such biases in block 1, participants were warned of the importance of digital forms for PLWH looking for dental care and that, if they wished to participate, they needed to respond to all of the digital forms to help these key populations and decrease the stigmatization of PLWH among dentists in Northern Brazil. A final source of bias was due to the fact that the current subgroup organization proved that the annual evaluation of each professional’s experience did not demonstrate as much effectiveness as expected; however, it could be observed that their knowledge increased after 5 years of professional experience, so more studies using a different subgroup organization every 5 years are necessary.

## 5. Conclusions

In conclusion, the present study observed and reported important data about the dental care of PLWH in the northern region of Brazil, exposing the lack of knowledge of professional dentists in this area, as well as their stigmas and prejudice about PLWH dental care. To improve dental treatment for PLWH, dentists may search for more classes or educational knowledge regarding HIV or patients affected by it and their families, which might increase the support needed by this key population. Therefore, with all the presented results, it is now possible for the Brazilian government, colleges, universities and the Brazilian Dental Council to create new public health strategies and to increase the dental treatment of key populations affected by HIV.

## Figures and Tables

**Figure 1 ijerph-20-06847-f001:**
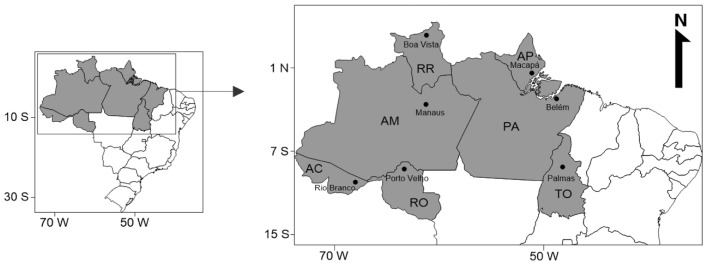
Geographic locations of all Northern Brazilian states and their capital municipalities.

**Table 1 ijerph-20-06847-t001:** Demographic and epidemiological profile of participant dentists from Northern Brazilian states.

Characteristics	Total n = 381 (100%)	G1 Group (≤1 Year) N = 51 (13.4%)	G2 Group (1 Years) N = 28 (7.3%)	G3 Group (2 Years) N = 37 (9.7%)	G4 Group (3 Years) N = 46 (12.1%)	G5 Group (4 Years) N = 35 (9.2%)	G6 Group (5 Years) N = 28 (7.3%)	G7 Group (≥5 Years) N = 156 (40.9%)	*p* Value
Gender *									
Male	188 (49.3%)	27 (53%)	15 (53.5%)	23 (62.2%)	19 (41.3%)	11 (31.4%)	13 (46.4%)	80 (51.2%)	0.1787
Female	193 (50.7%)	24 (47%)	13 (46.5%)	14 (37.8%)	27 (58.7%)	24 (68.6%)	15 (53.6%)	76 (48.8%)	
Age range (years) *									
22–29	164 (43%)	43 (84.3%)	27 (96.4%)	34 (92%)	27 (58.7%)	17 (48.5%)	8 (28.5%)	8 (5.1%)	<0.0001
30–39	117 (30.7%)	8 (15.7%)	1 (3.6%)	3 (8%)	14 (30.4%)	16 (45.7%)	16 (57.3%)	59 (37.8%)	
40–49	67 (17.5%)	-	-	-	5 (10.9%)	2 (5.8%)	4 (14.2%)	56 (36%)	
50–≥59	33 (8.8%)	-	-	-	-	-	-	33 (21.1%)	
State in which work *									
Pará	140 (36.7%)	29 (56.8%)	17 (60.7%)	11 (29.7%)	12 (26%)	3 (8.5%)	6 (21.4%)	62 (39.7%)	<0.0001
Amazonas	41 (10.8%)	5 (10%)	4 (14.3%)	4 (10.8%)	7 (15.2%)	3 (8.5%)	2 (7.2%)	16 (10.2%)	
Roraima	40 (10.5%)	1 (2.2%)	1 (3.6%)	2 (5.4%)	3 (6.5%)	7 (20%)	3 (10.7%)	23 (14.7%)	
Rondônia	40 (10.5%)	5 (10%)	2 (7.1%)	4 (10.8%)	2 (4.5%)	4 (11.4%)	6 (21.4%)	17 (11%)	
Amapá	40 (10.5%)	3 (5.5%)	1 (3.6%)	10 (27%)	4 (8.7%)	4 (11.4%)	2 (7.2%)	16 (10.2%)	
Acre	40 (10.5%)	3 (5.5%)	1 (3.6%)	5 (13.5%)	10 (21.7%)	8 (22.8%)	6 (21.4%)	7 (4.4%)	
Tocantins	40 (10.5%)	5 (10%)	2 (7.1%)	1 (2.8%)	8 (17.4%)	6 (17.4%)	3 (10.7%)	15 (9.8%)	
College type which graduated *									
Private	215 (56.4%)	40 (78.5%)	19 (67.8%)	25 (67.5%)	33 (71.7%)	27 (77.1%)	15 (53.5%)	56 (35.9%)	<0.0001
Public	166 (43.6%)	11 (21.5%)	9 (32.2%)	12 (32.5)	13 (28.3%)	8 (22.9%)	13 (46.5%)	100 (64.1%)	
Has completed postgraduation ^†^									
Specialization	273 (71.7%)	-	-	20 (54%)	39 (84.7%)	32 (91.4%)	26 (92.8%)	156 (100%)	<0.0001
Master’s degree	77 (20.2%)	-	-	-	3 (6.5%)	5 (14.2%)	6 (21.4%)	63 (40.3%)	
Doctorate degree	28 (7.3%)	-	-	-	-	-	2 (7.1%)	26 (16.6%)	
No postgraduation	94 (24.7%)	50 (98%)	26 (92.8%)	10 (27%)	6 (13%)	2 (5.8%)	-	-	
Other	15 (3.9%)	1 (2%)	2 (7.2%)	7 (19%)	-	1 (2.8%)	1 (3.5%)	3 (2%)	
Dental Specialty ^†^									
Pediatric dentistry	16 (4.2%)	-	-	3 (8%)	3 (6.5%)	2 (5.8%)	1 (3.5%)	7 (4.4%)	<0.0001
Orthodontics	57 (15%)	-	-	3 (8%)	6 (13%)	7 (20%)	4 (14.2%)	37 (23.7%)	
Facial and Jaw Orthopedics	8 (2.1%)	-	-	-	-	3 (8.5%)	2 (7.1%)	3 (2%)	
TMJ ^‡^ and Oral Pain	12 (3.1%)	-	-	1 (2.8%)	1 (2.1%)	1 (2.8%)	2 (7.1%)	7 (4.4%)	
Restorative dentistry	26 (6.8%)	-	-	-	4 (8.7%)	6 (17.1%)	2 (7.1%)	14 (9%)	
Geriatric dentistry	8 (2.1%)	-	-	-	3 (6.5%)	-	1 (3.5%)	4 (2.5%)	
Special care dentistry	10 (2.6%)	-	-	-	3 (6.5%)	1 (2.8%)	1 (3.5%)	5 (3.2%)	
Endodontics	44 (11.5%)	-	-	1 (2.8%)	6 (13%)	9 (25.7%)	5 (17.8%)	23 (14.7%)	
Periodontics	56 (14.7%)	-	-	4 (10.8%)	4 (8.7%)	3 (8.5%)	6 (21.4%)	39 (25%)	
Dental prosthesis	62 (16.3%)	-	-	3 (8%)	3 (6.5%)	9 (25.7%)	5 (17.8%)	42 (27%)	
Implantology	55 (14.4%)	-	-		6 (13%)	1 (2.8%)	5 (17.8%)	43 (27.5%)	
Maxillofacial Prosthesis	1 (0.3%)	-	-	-	-	-	-	1 (100%)	
Maxillofacial Surgery	28 (7.3%)	-	-		4 (8.7%)	1 (2.8%)	6 (21.4%)	17 (11%)	
Sports Dentistry	1 (0.3%)	-	-	-	1 (2.1%)	-	-	-	
Dental Facial Harmonization	23 (6%)	-	-	4 (10.8%)	4 (8.7%)	4 (11.4%)	5 (17.8%)	6 (3.8%)	
Pathology and Stomatology	20 (5.2%)	-	-	-	1 (2.1%)	1 (2.8%)	3 (10.7%)	15 (9.8%)	
Dental Radiology	12 (3.1%)	-	-	-	-	3 (8.5%)	-	9 (5.7%)	
Dentistry in public health	5 (1.3%)	-	-	1 (2.8%)	-	-	1 (3.5%)	3 (2%)	
Ongoing specialty	110 (28.9%)	51 (100%)	28 (100%)	17 (46%)	10 (21.7%)	4 (11.4%)	-	-	
Workplace ^†^									
Private	273 (71.7%)	20 (39.2%)	11 (39.4%)	26 (70.2%)	33 (71.7%)	29 (82.8%)	23 (82.1%)	131 (84%)	<0.0001
Public	167 (43.8%)	11 (21.6%)	13 (46.4%)	16 (27%)	25 (54.3%)	11 (31.4%)	13 (57.2%)	78 (50%)	
Currently not working	34 (8.9%)	20 (39.2%)	4 (14.2%)	1 (2.8%)	1 (2.1%)	-	-	8 (5.1%)	
Has already provided dental care to PLWH *									
Yes	225 (59.1%)	10 (19.6%)	14 (50%)	13 (35.1%)	27 (58.7%)	10 (28.5%)	22 (78.5%)	129 (82.7%)	<0.0001
No	67 (17.6%)	20 (39.2%)	3 (10.7%)	9 (24.3%)	5 (10.9%)	13 (37.1%)	4 (14.2%)	13 (8.3%)	
Maybe	89 (23.3%)	21 (58.8%)	11 (39.3%)	15 (40.6%)	14 (30.4%)	12 (34.4%)	2 (7.3%)	14 (9%)	
Has already had any class or lecture about PLWH care *									
Yes	210 (55.1%)	27 (53%)	14 (50%)	18 (48.7%)	19 (41.3%)	7 (20%)	17 (60.7%)	108 (69.3%)	<0.0001
No	171 (44.9%)	24 (47%)	14 (50%)	19 (51.3%)	27 (58.7%)	28 (80%)	11 (39.3%)	48 (30.7%)	

* dichotomous query; ^†^ multiple choice query; ^‡^ temporomandibular joint dysfunction.

**Table 2 ijerph-20-06847-t002:** Dentists’ answers about HIV/AIDS knowledge and PLWH dental management.

Queries	Total n = 381 (100%)	G1 Group (≤1 Year) N = 51 (13.4%)	G2 Group (1 Years) N = 28 (7.3%)	G3 Group (2 Years) N = 37 (9.7%)	G4 Group (3 Years) N = 46 (12.1%)	G5 Group (4 Years) N = 35 (9.2%)	G6 Group (5 Years) N = 28 (7.3%)	G7 Group (≥5 Years) N = 156 (40.9%)	*p* Value
Do you know the difference about what HIV ^¶^ and AIDS ^†^ is? *									
Yes (Apt)	105 (27.6%)	21 (41.1%)	9 (32%)	6 (16.3%)	13 (28.3%)	10 (28.6%)	8 (28.5%)	38 (24.3%)	0.2365
No (Inapt)	276 (72.4%)	30 (58.9%)	19 (68%)	31 (83.7%)	33 (71.7%)	25 (71.4%)	20 (71.5%)	118 (75.7%)	
Do you know HIV ^¶^ infection mechanism works? *									
Yes (Apt)	170 (44.6%)	21 (41.1%)	7 (25%)	19 (51.3%)	21 (45.6%)	9 (25.8%)	14 (50%)	79 (50.7%)	0.0382
No (Inapt)	211 (55.4%)	30 (58.9%)	21 (75%)	18 (48.7%)	25 (54.4%)	26 (74.2%)	14 (50%)	77 (49.3%)	
Do you know what HIV ^¶^ transmission mechanisms are? **									
Vertical transmission (Apt)	262 (68.8%)	32 (62.7%)	21 (75%)	23 (62.1%)	31 (67.4%)	29 (82.8%)	19 (67.8%)	107 (68.5%)	0.0013
Contamination by work accidents (Apt)	354 (92.9%)	48 (94.1%)	26 (92.8%)	34 (92%)	37 (80.4%)	32 (91.4%)	28 (100%)	149 (95.5%)	
Sharing hygiene products (Inapt)	65 (17.1%)	12 (23.5%)	7 (25%)	10 (27%)	8 (17.4%)	7 (20%)	8 (28.5%)	13 (8.3%)	
Via aerosol during dental care (Inapt)	136 (35.7%)	20 (39.2%)	10 (35.7%)	12 (32.4%)	17 (37%)	20 (57.1%)	15 (53.5%)	42 (27%)	
Contaminated blood transfusion (Apt)	353 (92.7%)	48 (94.1%)	25 (89.2%)	36 (97.3%)	42 (91.3%)	29 (82.8%)	26 (92.8%)	147 (94.2%)	
Unprotected sex (Apt)	356 (93.4%)	48 (94.1%)	27 (96.4%)	35 (94.5%)	42 (91.3%)	29 (82.8%)	25 (89.2%)	150 (96.1%)	
Personal contact by handshake or hug (Inapt)	18 (4.7%)	6 (11.7%)	2 (7.1%)	4 (10.8%)	2 (4.3%)	3 (8.5%)	1 (3.5%)	-	
Sharing needles to inject drugs (Apt)	343 (90%)	48 (94.1%)	23 (82.1%)	31 (83.7%)	37 (80.4%)	28 (80%)	25 (89.2%)	151 (96.8%)	
Sharing plates and cutlery (Inapt)	32 (8.4%)	6 (11.7%)	2 (7.1%)	2 (5.4%)	5 (10.8%)	8 (22.8%)	4 (14.2%)	5 (3.2%)	
Contact with mucous membranes, wounds or body fluids (Inapt)	113 (29.7%)	24 (47%)	5 (17.8%)	12 (32.4%)	14 (30.4%)	19 (54.2%)	10 (35.7%)	29 (18.5%)	
Do you know what the most common oral lesions are in PLWH ^‡^? **									
Xerostomia (Inapt)	69 (18.1%)	13 (25.4%)	5 (17.8%)	5 (13.5%)	10 (21.7%)	6 (17.1%)	5 (17.8%)	25 (16%)	<0.0001
Aphthous ulcerations (Inapt)	150 (39.4%)	26 (51%)	16 (57.1%)	16 (43.2%)	18 (39.1%)	9 (25.8%)	10 (35.7%)	55 (35.2%)	
Condyloma (Inapt)	69 (18.1%)	10 (19.6%)	4 (14.2%)	11 (29.8%)	5 (10.8%)	3 (8.5%)	5 (17.8%)	31 (19.8%)	
Necrotizing periodontitis (Inapt)	155 (40.7%)	34 (66.6%)	14 (50%)	15 (40.6%)	20 (43.4%)	17 (48.5%)	7 (25%)	48 (30.7%)	
Oral candidiasis (Inapt)	207 (54.3%)	35 (68.6%)	17 (60.7%)	23 (62.1%)	24 (52.1%)	17 (48.5%)	21 (75%)	70 (44.8%)	
Herpetic gingivostomatitis (Inapt)	116 (30.4%)	15 (29.4%)	13 (46.4%)	11 (29.8%)	16 (34.7%)	10 (28.6%)	5 (17.8%)	46 (29.4%)	
Necrotizing gingivitis (Inapt)	178 (46.7%)	34 (66.6%)	12 (42.8%)	21 (56.7%)	23 (50%)	16 (45.7%)	8 (28.5%)	64 (41%)	
Hairy leukoplakia (Inapt)	102 (26.8%)	10 (19.6%)	5 (17.8%)	11 (29.8%)	16 (34.7%)	11 (31.4%)	6 (21.4%)	43 (27.5%)	
Linear Gingival Erythema (Inapt)	70 (18.4%)	8 (15.6%)	8 (28.5%)	5 (13.5%)	9 (19.5%)	10 (28.6%)	6 (21.4%)	24 (15.3%)	
Necrotizing stomatitis (Inapt)	114 (29.9%)	23 (45%)	5 (17.8%)	13 (36%)	23 (50%)	16 (45.7%)	8 (28.5%)	26 (16.6%)	
Focal epithelial hyperplasia (Inapt)	44 (11.5%)	3 (5.8%)	5 (17.8%)	4 (10.8%)	5 (10.8%)	7 (20%)	2 (7.1%)	18 (11.5%)	
Kaposi’s sarcoma (Inapt)	182 (47.8%)	26 (51%)	16 (57.1%)	15 (40.6%)	22 (47.8%)	14 (40%)	25 (89.2%)	64 (41%)	
Angular cheilitis (Inapt)	67 (17.6%)	14 (27.4%)	3 (10.7%)	7 (19%)	8 (17.4%)	5 (14.2%%)	3 (10.7%)	27 (17.3%)	
All alternatives above (Apt)	114 (29.9%)	-	3 (10.7%)	6 (16.3%)	12 (26%)	9 (25.8%)	12 (42.8%)	72 (46.1%)	
Do you know what the most common signs and symptoms of HIV ^¶^ infection are? **									
Prolonged flu (Inapt)	157 (41.2%)	27 (53%)	14 (50%)	18 (48.7%)	19 (41.3%)	17 (48.5%)	12 (42.8%)	50 (32%)	0.1844
Fever (Inapt)	118 (31%)	19 (37.2%)	9 (32%)	14 (37.8%)	15 (32.6%)	13 (37.1%)	10 (35.7%)	38 (24.3%)	
Dry cough (Inapt)	79 (20.7%)	13 (25.5%)	8 (28.5%)	8 (21.7%)	12 (26%)	8 (22.8%)	5 (17.8%)	25 (16%)	
Malaise (Inapt)	117 (30.7%)	19 (37.2%)	6 (21.4%)	8 (21.7%)	18 (39.1%)	10 (28.6%)	8 (28.5%)	48 (30.7%)	
Sore throat (Inapt)	56 (14.7%)	13 (25.5%)	4 (14.2%)	7 (19%)	4 (8.7%)	2 (5.7%)	7 (25%)	19 (12.1%)	
Nausea (Inapt)	83 (21.8%)	18 (35.3%)	6 (21.4%)	10 (27%)	9 (19.5%)	6 (17.1%)	9 (32.1%)	25 (16%)	
Vomit (Inapt)	80 (21%)	21 (41.1%)	4 (14.2%)	10 (27%)	11 (24%)	5 (14.2%%)	8 (28.5%)	21 (13.4%)	
Lymph node enlargement (Inapt)	156 (40.9%)	18 (35.3%)	14 (50%)	19 (51.3%)	20 (43.4%)	10 (28.6%)	8 (28.5%)	67 (43%)	
Fatigue (Inapt)	111 (29.1%)	18 (35.3%)	9 (32%)	9 (24.3%)	15 (32.6%)	8 (22.8%)	6 (21.4%)	46 (29.4%)	
Constant weight loss (Inapt)	208 (54.6%)	35 (68.6%)	16 (57.1%)	23 (62.1%)	22 (47.8%)	17 (48.5%)	20 (71.5%)	75 (48%)	
Night sweats (Inapt)	66 (17.3%)	7 (13.7%)	7 (25%)	6 (16.3%)	7 (15.2%)	8 (22.8%)	7 (25%)	24 (15.3%)	
Persistent diarrhea (Inapt)	134 (35.2%)	17 (33.3%)	13 (46.4%)	12 (32.4%)	17 (37%)	16 (45.7%)	11 (7%)	48 (30.7%)	
Appetite loss (Inapt)	144 (37.8%)	27 (53%)	13 (46.4%)	18 (48.7%)	19 (41.3%)	8 (22.8%)	10 (35.7%)	49 (31.4%)	
All alternatives above (Apt)	138 (36.2%)	6 (11.7%)	6 (21.4%)	9 (24.3%)	16 (34.7%)	8 (22.8%)	12 (42.8%)	81 (52%)	
Do you know how to request HIV medical laboratory exams? *									
Yes (Apt)	173 (45.4%)	14 (27.4%)	7 (25%)	17 (46%)	17 (37%)	8 (22.8%)	13 (46.4%)	97 (62%)	<0.0001
No (Inapt)	208 (54.6%)	37 (72.6%)	21 (75%)	20 (54%)	29 (63%)	27 (77.2%)	15 (53.6%)	59 (38%)	
Do you how to proceed after an occupational exposure accident during PLWH ^‡^ dental management? *									
Yes (Apt)	186 (48.8%)	19 (37.3%)	5 (17.9%)	22 (59.4%)	14 (30.4%)	8 (22.8%)	15 (53.6%)	103 (66%)	<0.0001
No (Inapt)	195 (51.2%)	32 (62.7%)	23 (82.1%)	15 (40.6%)	32 (69.6%)	27 (77.2%)	13 (46.4%)	53 (34%)	
Do you know what HIV ^¶^ window period is? *									
Yes (Apt)	197 (51.7%)	20 (39.2%)	10 (35.8%)	22 (59.4%)	17 (37%)	11 (31.4%)	14 (50%)	103 (66%)	<0.0001
No (Inapt)	184 (48.3%)	31 (60.8%)	18 (64.2%)	15 (40.6%)	29 (63%)	24 (68.6%)	14 (50%)	53 (34%)	
Do you know what biosafety protocol is better for PLWH ^‡^ dental care and material sterilization? *									
Yes (Inapt)	130 (34.1%)	27 (53%)	12 (43%)	13 (36%)	12 (26%)	12 (34.2%)	8 (28.5%)	46 (29.4%)	0.0710
No (Apt)	251 (65.9%)	24 (47%)	16 (57%)	24 (64%)	34 (74%)	23 (65.8%)	20 (71.5%)	110 (70.6%)	
Do you know about ART ^#^ medications and interactions with dental prescription drugs? *									
Yes (Apt)	165 (43.3%)	4 (7.9%)	8 (28.6%)	8 (21.7%)	20 (43.5%)	11 (31.4%)	10 (35.8%)	104 (66.6%)	<0.0001
No (Inapt)	216 (56.7%)	47 (92.1%)	20 (71.4%)	29 (78.3%)	26 (56.5%)	24 (68.6%)	18 (64.2%)	52 (33.4%)	
Do you do any kind of changes in your non-invasive dental care protocol to treat PLWH ^‡^? *									
Yes (Inapt)	146 (38.3%)	38 (74.5%)	11 (39.2%)	21 (56.7%)	26 (56.5%)	13 (38%)	11 (39.3%)	26 (16.7%)	<0.0001
No (Apt)	235 (61.7%)	13 (25.5%)	17 (60.8%)	16 (43.4%)	20 (43.5%)	22 (62%)	17 (60.7%)	130 (83.3%)	
Do you do any kind of changes in your invasive dental care protocol to treat PLWH ^‡^? *									
Yes (Inapt)	135 (35.4%)	29 (56.8%)	16 (57%)	19 (51.3%)	17 (37%)	13 (38%)	10 (35.8%)	31 (19.9%)	<0.0001
No (Apt)	246 (64.6%)	22 (43.2%)	12 (43%)	18 (48.7%)	29 (63%)	22 (62%)	18 (64.2%)	125 (80.1%)	
Do you have any kind of fears or stigma regarding PLWH ^‡^ dental care? *									
Yes (Inapt)	295 (77.4%)	42 (82.3%)	18 (64.2%)	26 (70.2%)	30 (65.2%)	24 (68.6%)	22 (78.5%)	133 (85.2%)	0.0207
No (Apt)	86 (22.6%)	9 (17.7%)	10 (35.8%)	11 (29.8%)	16 (34.8%)	11 (31.4%)	6 (21.5%)	23 (14.8%)	

* dichotomous query; ** multiple choice query; ^¶^ HIV: human immunodeficiency virus; ^†^ AIDS: acquired immunodeficiency syndrome; ^‡^ PLWH: people living with HIV; ^#^ antiretroviral therapy.

**Table 3 ijerph-20-06847-t003:** Knowledge variation of dentists before and after queries about PLWH oral care.

	Knowledge Level				
Moments (Classification)	Sufficient n = 532 (100%)	CI 95%	Insufficient n = 230 (100%)	CI 95%	*p*-Value *
At the beginning of the form (knowledge informed)	215 (40.5%)	0.362 (36.2%)–0.446 (44.6%)	166 (72.1%)	0.664 (66.4%)–0.780 (78.0%)	<0.0001
At the end of the form (knowledge informed)	317 (59.5%)	0.554 (55.4%)–0.638 (63.8%)	64 (27.9%)	0.220 (22.0%)–0.336 (33.6%)	
Knowledge improvement demonstrated	102 (19.1%)	0.158 (15.8%)–0.225 (22.5%)	102 (44.3%)	0.379 (37.9%)–0.508 (50.8%)	

**Table 4 ijerph-20-06847-t004:** Knowledge variation of dentists before and after queries about PLWH oral care by each group.

Moments (Classification)	Total n =762 (100%)	G1 Group (≤1 Year) N = 102 (13.3%)	G2 Group (1 Years) N = 56 (7.3%)	G3 Group (2 Years) N = 74 (9.7%)	G4 Group (3 Years) N = 92 (12%)	G5 Group (4 Years) N = 70 (9.2%)	G6 Group (5 Years) N = 56 (7.3%)	G7 Group (≥5 Years) N = 312 (41%)	*p* Value
Inapt scale of self-informed knowledge at the beginning of the form	166 (21.7%)	35 (34.3%)	20 (35.7%)	20 (27%)	21 (22.8%)	24 (34.2%)	12 (21.5%)	34 (11%)	<0.0001
Apt scale of self-informed knowledge at the beginning of the form	215 (28.2%)	16 (15.8%)	8 (14.4%)	17 (23%)	25 (27.1%)	11 (15.7%)	16 (28.5%)	122 (39.1%)	
Inapt scale of self-informed knowledge at the end of the form	64 (8.4%)	22 (21.5%)	12 (21.4%)	5 (6.8%)	6 (6.5%)	3 (4.4%)	0 (0%)	16 (5.1%)	<0.0001
Apt scale of self-informed knowledge at the end of the form	317 (41.7%)	29 (28.4%)	16 (28.5%)	32 (43.2%)	40 (43.6%)	32 (45.7%)	28 (50%)	140 (44.8%)	

## Data Availability

All data referred to this study is available on the manuscript and supplementary material attached to this manuscript.

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
