# Peer review of "Digital Assessment of the Knowledge, Attitudes and Preparedness of Dentists towards Providing Dental Treatment to People Living with HIV in Northern Brazil"

_ijerph, 2023, doi:10.3390/ijerph20196847_

Round 1

Reviewer 1 Report (Previous Reviewer 3)

in the methodology: the questionairres  to measure knowledge  of dentist Is it already validated ? because it is an instrument to measure abstract variables I think it should be validated 

if it is already validated  please add in the article

in the abstract section:  result ..the sentence G7 groups.... until the sentence were most cited ...

I think the author should  revised this sentence to make it more clear and easy to understand

Author Response

Reply to reviewer #1

1. Concern of the reviewer • In the methodology: the questionnaire to measure knowledge of dentist Is it already validated ? because it is an instrument to measure abstract variables I think it should be validated if it is already validated please add in the article. Our response:               Dear Reviewer #1, we appreciate your concerning regarding the questionnaire validation, as mentioned in our paper on page 4, lines: 155 to 156 our questionnaire was used and validated in a prior pilot study which is used as reference. Although I took the liberty to attach the link to the pilot study paper that was published. Link: https://www.mdpi.com/1660-4601/19/9/5055. 2. Concern of the reviewer• Comments on the Quality of English Language in the abstract section: result ..the sentence G7 groups.... until the sentence were most cited ... I think the author should revised this sentence to make it clearer and easier to understand. Our response:               Dear Reviewer #2, we appreciate your comment and the abstract was revised.

Revised text:Page 1, lines 23-40, “Background: People living with HIV (PLWH) continues to face stigma and discrimination during dental treatment in Brazil. This study aimed to describe the sociodemographic, educational and professional characteristics of dentists working in northern Brazilian region, as well as identifying the level of knowledge of these health professionals in relation to the care and management of PLWH using a digital form. Methods: This cross-sectional study was population-based in dentists was conducted between July to December 2021, a total of 396 dentists were invited using “snowball” sampling technique received an anonymous digital form (Google® Forms Platform) composed of four blocks dichotomous and multiple-choice questions. After sign the informed consent term, dentists were divided into seven groups according their bachelor’s degree in dentistry completed time (in years). A total of 25 questions referred to sociodemographic, educational and professional data, also knowledge about dental care of PLWH were asked and all data were presented as descriptive percentages and then analyzed using the Kappa test. Results: The average age was 31.9 years and Pará and Amazonas states presented better knowledge about dental care of PLWH, Dental prosthesis (62/381 - 16.3%); Orthodontics (57/381 - 15%) and Periodontics (56/381 - 14.7%) were the most cited dental specialties and demographic, professional and epidemiological characteristics showed a statistically significant regarding knowledge (< 0.0001). Conclusions: The knowledge of dentists in northern Brazil is partially high, although there are some lack of knowledge about dental care of PLWH were evidenciated.”

Reviewer 2 Report (Previous Reviewer 2)

The article was improved sufficiently. Many of the previous comments have been addressed; however, some comments remain.

General note:

English - proofreading is still needed.

Abstract

The abstract is too long (please, check authors guidelines). Also G7 is mentioned without an explanation what this G7 entails.

Materials and methods

The same commentary.

The dentists were divided into seven subgroups according their self-declared academic completion period time in years, so groups were organised as: G1 group: ≤1 year 163 (control group); G2 group: 1 years; G3 group: 2 years; G4 group: 3 years; G5 group: 4 years; G6 group: 5 years; G7 group: ≥5 years such division is questionable. I understand, that dentists who work for only 1 year differ from those who work at list 5 years, but I do not think that comparisons between those having 2 or 3 or 4 years of experience are feasible. Any statistical differences shown in such comparisons are likely to be random. I suggest to rearrange the groups (something like less than 5 years, 6-10 years, 11-15 and so on) as group 7 is the largest one. This at least can be discussed.  The justification for such division can be explained in the discussion section. Also, possible bias due to not having more experienced dentists in separate groups should be mentioned in the limitations section.

Moderate editing of English language required.

Author Response

Reply to reviewer #2

1. Concern of the reviewer              • General note: English - proofreading is still needed. Our response:               Dear Reviewer #2, we appreciate your concern. The text was carefully revised by a professional translator in english. 2. Concern of the reviewer• The abstract is too long (please, check authors guidelines). Also G7 is mentioned without an explanation what this G7 entails. Our response:               Dear Reviewer #2, we appreciate your comment and the abstract was revised.

Revised text:Page 1, lines 23-40, “Background: People living with HIV (PLWH) continues to face stigma and discrimination during dental treatment in Brazil. This study aimed to describe the sociodemographic, educational and professional characteristics of dentists working in northern Brazilian region, as well as identifying the level of knowledge of these health professionals in relation to the care and management of PLWH using a digital form. Methods: This cross-sectional study was population-based in dentists was conducted between July to December 2021, a total of 396 dentists were invited using “snowball” sampling technique received an anonymous digital form (Google® Forms Platform) composed of four blocks dichotomous and multiple-choice questions. After sign the informed consent term, dentists were divided into seven groups according their bachelor’s degree in dentistry completed time (in years). A total of 25 questions referred to sociodemographic, educational and professional data, also knowledge about dental care of PLWH were asked and all data were presented as descriptive percentages and then analyzed using the Kappa test. Results: The average age was 31.9 years and Pará and Amazonas states presented better knowledge about dental care of PLWH, Dental prosthesis (62/381 - 16.3%); Orthodontics (57/381 - 15%) and Periodontics (56/381 - 14.7%) were the most cited dental specialties and demographic, professional and epidemiological characteristics showed a statistically significant regarding knowledge (< 0.0001). Conclusions: The knowledge of dentists in northern Brazil is partially high, although there are some lack of knowledge about dental care of PLWH were evidenciated.”

  1. Concern of the reviewer
  • Materials and methods: The dentists were divided into seven subgroups according their self-declared academic completion period time in years, so groups were organized as: G1 group: ≤1 year 163 (control group); G2 group: 1 years; G3 group: 2 years; G4 group: 3 years; G5 group: 4 years; G6 group: 5 years; G7 group: ≥5 years such division is questionable. I understand, that dentists who work for only 1 year differ from those who work at list 5 years, but I do not think that comparisons between those having 2 or 3 or 4 years of experience are feasible. Any statistical differences shown in such comparisons are likely to be random. I suggest to rearrange the groups (something like less than 5 years, 6-10 years, 11-15 and so on) as group 7 is the largest one. This at least can be discussed.  The justification for such division can be explained in the discussion section. Also, possible bias due to not having more experienced dentists in separate groups should be mentioned in the limitations section.

Our response: Dear Reviewer #2, we appreciate your suggestion and concern. Regarding subgroup division we used this division in order to understand in each professional experience year, dentists may increase their knowledge about HIV subject and according to our results the knowledge seems to increase in every 5 years, as you suggested in the above comment. Although, your suggestion to rearrange the groups in less than 5 years, 6-10 years, 11-15 and so on is major excellent by observing our results, we will use your suggestion in our next paper about this topic, mainly based in your suggestion and in our results. This is due to our currently paper is finished, and the results presented have important informations to the public written and organized as it is, we hope that true and honest response might satisfy you and now you may suggest our paper for publication. Also, in study limitations we warned readers upon this bias in our study and in the future your suggestion will be used. Revised text:Page 18, lines 461-465, “A final bias is that the current subgroup organization proved that annual evaluation of each professional experience did not demonstrated as much effectiveness as expected, however, it could be observed that knowledge increased occurred after 5 years of professional experience, so more studies using a different subgroup organization in every 5 years are necessary.”

This manuscript is a resubmission of an earlier submission. The following is a list of the peer review reports and author responses from that submission.

Round 1

Reviewer 1 Report

Thank you for the opportunity to review this article. Original and interesting article even if restricted to one country and therefore could not be of global interest. I recommend reformatting the tables, which are difficult to understand as they are.

Author Response

Reply to reviewer #1

Concern of the reviewer

Thank you for the opportunity to review this article. Original and interesting article even if restricted to one country and therefore could not be of global interest. I recommend reformatting the tables, which are difficult to understand as they are.

Our response:              

Dear Reviewer #1, we appreciate your suggestion and all tables were revised according to your suggestion. Although this paper seems to be restricted to just one country, a few papers in literature presented similar results about lack of knowledge among dental students or dentists, therefore our paper is not alone in its results and also corroborates others results, which could be used worldwide to improve knowledge about the topic. 

Reviewer 2 Report

The paper “Digital assessment of knowledge, attitude and preparedness of dentists towards people living with HIV in northern Brazil” reports on the survey conducted among Brazilian dentists of the Northern region. The study is well-planned and thoroughly described. The topic is really important. However, there are some comments that should be addressed.

General note:

English - extensive editing is needed to some parts of the manuscript.

Title: Digital assessment of knowledge, attitude and preparedness of dentists towards people living with HIV in northern Brazil - I suggest to rephrase: “…towards providing dental treatment to patients…”

Introduction

Lines 76-81 this paragraph is unnecessary.

Introduction may be shortened (mainly in terms of general information about AIDS and so on), but more data are needed about previously published surveys on the topic.

Materials and methods

The dentists were divided into seven subgroups according their self-declared academic completion period time in years, so groups were organised as: G1 group: ≤1 year 163 (control group); G2 group: 1 years; G3 group: 2 years; G4 group: 3 years; G5 group: 4 years; G6 group: 5 years; G7 group: ≥5 years such division is questionable. I understand, that dentists who work for only 1 year differ from those who work at list 5 years, but I do not think that comparisons between those having 2 or 3 or 4 years of experience are feasible. Any statistical differences shown in such comparisons are likely to be random. I suggest to rearrange the groups (something like less than 5 years, 6-10 years, 11-15 and so on) as group 7 is the largest one.

202 BioEstat 5.0 program: please, specify company name, city, country.

As it can be assumed from the table, the questions were mainly in the form “Do you know…?” . For example “Do you know the difference about what HIV and AIDS is?” Why did you chose this type of questions instead of actually checking the knowledge? For example “What is the difference between HIV and AIDS?” with different possible answers? With questions like “Do you know…?”, a risk of response bias is quite high. This should be discussed in the limitations.

Please, consider assessing the overall knowledge and attitude scores (number of correct/positive answers) and comparing it between the study groups.

Results

Table 2 is not necessary. It should be removed or moved to the supplement section. Another suggestion is to present this information partially as pie-charts. Also, I such a detailed description of demographics in lines 234-254 is unnecessary.

Conclusion

In conclusion the major study finding should be repeated in general, but more detailed than it is now. I think that the part “the present study observed and reported unknown facts and data” is unnecessary.

Author Response

Reply to reviewer #2

Concern of the reviewer          

  • General comment: English - extensive editing is needed to some parts of the manuscript. 

Our response:              

Dear Reviewer #2, we appreciate your concern. The text was carefully revised by a professional translator in english. 

2. Concern of the reviewer

Title: Digital assessment of knowledge, attitude and preparedness of dentists towards people living with HIV in northern Brazil - I suggest to rephrase: “…towards providing dental treatment to patients… 

Our response:              

Dear Reviewer #2, we appreciate your comment and the title was revised.

 Revised text:Page 1, lines 2-4, “Digital assessment of knowledge, attitude and preparedness of dentists towards providing dental treatment to people living with HIV in northern Brazil.”

3. Concern of the reviewer

  • Introduction: Lines 76-81 this paragraph is unnecessary.

Our response: Dear Reviewer #2, we appreciate your suggestion and concern. The text was carefully added in methods section, now reading the manuscript we agreed that this paragraph is unnecessary in introduction, although we moved to methods because we would like to base our results in these geographic informations. 

Revised text:Page 3, lines 106-115, Brazilian northern region is composed by seven states: Acre (capital: Rio Branco), Amapá (capital: Macapá), Amazonas (capital: Manaus), Pará (capital: Belém), Rondônia (capital: Porto Velho), Roraima (capital: Boa Vista) and Tocantins (capital: Palmas) [7,18]. The northern region is characterized by a large amazon rain forest territory and a variety of watersheds which represents a territorial area of 3.853. 676 km², which is almost 45% of Brazilian national territory with total population of 18.906.962 people, making northern region least inhabited region in Brazil. Besides geographical isolation, Brazilian northern region has a highway connectivity limitation, lower industrialization areas, decrease goods productivity, higher poverty rates and less structured public health services causing a difficulty to access specialized AIDS treatment [18-20].”  

4. Concern of the reviewer

  • Introduction: may be shortened (mainly in terms of general information about AIDS and so on), but more data are needed about previously published surveys on the topic.

Our response: Dear Reviewer #2, we appreciate your suggestion and concern. The text was carefully revised.

Revised text:Page 2, lines 49-93,In Brazil public or private health services availability versus population's access to health services always has been heterogeneous, mainly, due to regional inequalities as social, cultural, economic, demographic, geographic and its resolution is a complex challenge to Brazilian government [1]. Specifically, there are differences between health services availability and population access, cause in some regions such as Brazilian northern region, the offer or availability of public and/or private health services such as dentistry, to northern population, may be compromised mainly by economic and geographic barriers, which directly impacts dental health decreasing life quality, especially, among key populations to HIV infection [2,3]. According Fonseca et al. [4] several oral manifestations like oral candidiasis, necrotizing gingivitis or periodontitis, Kaposi sarcoma and leukoplakia are commonly found among PLWH serving as diagnosis or indications for HIV infection or determine the individual's stage of HIV infection demonstrating the major importance of dentists in specialized AIDS treatment and HIV infection control [5]. Unfortunately, some studies reported certain unwillingness by dentists and dental students to treat properly PLWH, which has been associated with stigma over HIV transmission during dental management or inadequate knowledge of HIV infection process resulting in fear, phobia or stigma of HIV contamination, demonstrating that dentist and dental students require specific education and training in specialized AIDS treatment and HIV infection control by universities, health institutes and federal government [6-8]. Historically, HIV infection and key populations are strongly marked by stigma and prejudice by families, society and even health workers, like dentists, that may lead HIV-infected individuals to social isolation, physical, dental and mental health problems [9]. Interestingly, about dental care or treatment, previous studies demonstrated that PLWH were unwilling to seek for dental treatment and, also, hesitate to disclose their HIV status to dentists, as a way to prevent stigmatization. In literature, it has been reported that dentists with low knowledge level about HIV infection oftenly refuse to treat PLWH, as well is described that lack of knowledge in dental workers, dental students and dentists can decrease oral health and life quality among PLWH due to oral diseases as periodontitis higher presence [10-12]. Worldwide, a diversity of forms in different studies were used to measure knowledge, attitude and willingness to dental treatment of PLWH [13-16]. Despite different evaluation methods the majority of studies showed that the lower willingness to dental treat HIV infected patients among dental students or dentists might be linked to their knowledge about HIV infection-disease aspects, recognition of oral manifestations, and understanding of the modes of transmission, these results may indicate that HIV/AIDS stigma among dentists and dental students are directly related to lower knowledge levels which leads to higher levels of stigmatizing or negative willingness [15,16]. Therefore, this lack of knowledge, fear of contracting HIV, HIV/AIDS multidisciplinary treatment remains a major dental health problem to HIV-infected individuals or key population in countries like China, India, Indonesia and Brazil, mainly, in Brazilian northern region [17]. Thus, this study described the sociodemographic, educational and professional characteristics of dentists working in northern Brazilian region, as well as identifying the level of knowledge of these health professionals in relation to the care and management of PLWH using a digital form.”  

5. Concern of the reviewer

  • Materials and methods: The dentists were divided into seven subgroups according their self-declared academic completion period time in years, so groups were organised as: G1 group: ≤1 year 163 (control group); G2 group: 1 years; G3 group: 2 years; G4 group: 3 years; G5 group: 4 years; G6 group: 5 years; G7 group: ≥5 years such division is questionable. I understand, that dentists who work for only 1 year differ from those who work at list 5 years, but I do not think that comparisons between those having 2 or 3 or 4 years of experience are feasible. Any statistical differences shown in such comparisons are likely to be random. I suggest to rearrange the groups (something like less than 5 years, 6-10 years, 11-15 and so on) as group 7 is the largest one.

Our response: Dear Reviewer #2, we appreciate your suggestion and concern. The results were presented like this so we could identify the knowledge curve among dentists and also identify where the knowledge  improvement is needed, if during dental college or after 5 years. Yes we agreed that group seven is by far the largest one among the participants, but we do not impose any kind of limits among groups participants because this digital form was conducted during the COVID-19 pandemic and also free and voluntary participation, therefore more experienced dentists participated more than less experienced dentists. 

6. Concern of the reviewer

  • Materials and methods: 202 BioEstat 5.0 program: please, specify company name, city, country.

Our response: Dear Reviewer #2, we appreciate your suggestion and concern. The text was carefully added in methods section. 

Revised text:Page 4, lines 183-184, then all statistical procedures were run in the BioEstat 5.0 program (Informer technologies inc., Manaus, Brazil).”  

7. Concern of the reviewer

  • Results: As it can be assumed from the table, the questions were mainly in the form “Do you know…?” . For example “Do you know the difference about what HIV and AIDS is?” Why did you chose this type of questions instead of actually checking the knowledge? For example “What is the difference between HIV and AIDS?” with different possible answers? With questions like “Do you know…?”, a risk of response bias is quite high. This should be discussed in the limitations.

Our response: Dear Reviewer #2, we appreciate your suggestion and concern. The questions were elaborated using a few studies in literature + pilot study + regional common doubts among dentists in Brazil and in the attached linkes below. Believe me even nowadays, in some Brazilian areas HIV and AIDS are the same thing, so we would like to use the digital form to clarify these doubts. Also following the HIV questions there was a discursive question asking for the participants to explain this difference and that’s how we measure if participants were right or wrong in there’s answers. Otherwise I do not know your familiarity with digital form studies, however a common bias is the difficulties to people answer discursive question  or massive forms, and this problem was corroborated during our pilot study in the process of form validation, so the best way to do our study was organizing our form like that. We count on your comprehension to help accept our paper. 1-      https://research.aimultiple.com/online-survey-challenges/2-      https://www.proprofssurvey.com/blog/advantages-disadvantages-of-online-surveys/  

8. Concern of the reviewer

  • Results: Please, consider assessing the overall knowledge and attitude scores (number of correct/positive answers) and comparing it between the study groups.

Our response: Dear Reviewer #2, we appreciate your suggestion and concern. The results were already compared between groups. 

9. Concern of the reviewer

  • Results: Table 2 is not necessary. It should be removed or moved to the supplement section. Another suggestion is to present this information partially as pie-charts. Also, I such a detailed description of demographics in lines 234-254 is unnecessary.

Our response: Dear Reviewer #2, we appreciate your suggestion and concern. The text was carefully revised and table 2 was removed, also the paragraph suggested by you was removed. 

10. Concern of the reviewer

  • Conclusion: In conclusion the major study finding should be repeated in general, but more detailed than it is now. I think that the part “the present study observed and reported unknown facts and data” is unnecessary.

Our response: Dear Reviewer #2, we appreciate your suggestion and concern. The text was carefully revised.  

Revised text:Page 18, lines 470-478, In conclusion, the present study observed and reported important data about dental care of PLWH in Brazilian northern region exposing the lack of knowledge of professional dentists of this area, but also their stigmas and prejudice about PLWH dental care. In order to improve dental treatment towards PLWH, dentists may search for more classes or education knowledge regarding HIV or HIV-infected patients and their families, which might increase support needed by this key population. Therefore with all the presented results now is possible to Brazilian governement, colleges, universities and Brazilian dentistry national council to create new public health strategies and to increase dental treatment of key population infected by HIV.”  

Reviewer 3 Report

Methods section: 

1. Dentists included in this study were divided into 7 subgroups, is it possible to simplify the subgroups? it will be easier to conclude the result.

Please explain why the author divided the dentists into 7 subgroups.

2. The author uses digital block form. Block 4 is based on several questions to measure knowledge. Please add information on whether this questionnaire has already been validated

Result section

1. Table 2  I don't think this table is important for the reader of the journal.

I think it should be better if it is simplified

2. Table 3. I think it should be better to classify the item of questions into subgroups to measure the specific outcome

Discussion section:

Please add a discussion regarding possible bias risk  in this study and how the author overcome this bias

Author Response

Reply to reviewer #3

1. Concern of the reviewer              

• Methods section: Dentists included in this study were divided into 7 subgroups, is it possible to simplify the subgroups? it will be easier to conclude the result. Please explain why the author divided the dentists into 7 subgroups. 

Our response: Dear Reviewer #3, we appreciate your suggestion. The text was carefully added to explain why we used year of experience as method to divide dentists in 7 subgroups. 

Revised text:Page 4, lines 155-158, This division occurred because we would like to understand if more experienced dentists had more knowledge than less experienced dentists, and to understand which year of dental clinical experience HIV knowledge among dentists is well established.” 

2. Concern of the reviewer           

• Methods section: The author uses digital block form. Block 4 is based on several questions to measure knowledge. Please add information on whether this questionnaire has already been validated.

Our response: Dear Reviewer #3, we appreciate your concern. The digital form was previously used and validated by a pilot study, then published in 2022. The pilot study is available in: https://www.ncbi.nlm.nih.gov/pmc/articles/PMC9103845/pdf/ijerph-19-05055.pdf. 

3. Concern of the reviewer             

• Result section: Table 2 I don't think this table is important for the reader of the journal. I think it should be better if it is simplified. 

Our response: Dear Reviewer #3, we appreciate your concern. The text was carefully revised and table 2 was removed, and the following paragraphs containing table 2 informations were also removed. 

4. Concern of the reviewer            

 • Result section: Table 3 I think it should be better to classify the item of questions into subgroups to measure the specific outcome

Our response: Dear Reviewer #3, we appreciate your concern. Although table 3 has been already classified according our subgroups, but were rewrote table 3 and text in order to improve text comprehension.

5. Concern of the reviewer             

• Discussion section: Please add a discussion regarding possible bias risk in this study and how the author overcome this bias. 

Our response: Dear Reviewer #3, we appreciate your concern. The text was carefully revised and added.

Revised text:Page 18, lines 463-471, While this larger study was successful in terms of the data collected and present data, it had certain limitations, such as restriction to Braziian northern region, only, showing northern characteristics only, the exclusion of individuals that did not have access to the internet or do not wanted to participate of our study may be a bias on ou rseults, also to finish soon the digital form, participants could answered quickly or participants did not disclosed correctly demographic information. In order to prevent such biases in block 1, participants were warned the importance of digital form for PLWH looking for dental care, and if they wanted to participated, they need to respond all digital form to help these key population and decrease stigmatization of PLWH among dentists in northern Brazil.”